# Edible Chitosan/Propolis Coatings and Their Effect on Ripening, Development of *Aspergillus flavus*, and Sensory Quality in Fig Fruit, during Controlled Storage

**DOI:** 10.3390/plants10010112

**Published:** 2021-01-07

**Authors:** Pablo F. Aparicio-García, Rosa I. Ventura-Aguilar, Juan C. del Río-García, Mónica Hernández-López, Dagoberto Guillén-Sánchez, Dolores A. Salazar-Piña, Margarita de L. Ramos-García, Silvia Bautista-Baños

**Affiliations:** 1Facultad de Nutrición, Universidad Autónoma del Estado de Morelos, Calle Iztaccihuatl S/N, Col. Los Volcanes, Cuernavaca, Morelos C.P. 62350, Mexico; lnpabloaparicio@gmail.com (P.F.A.-G.); azucena.salazar@uaem.mx (D.A.S.-P.); 2CONACYT-Centro de Desarrollo de Productos Bióticos, Instituto Politécnico Nacional, Carretera Yautepec-Jojutla, km. 6.8, San Isidro, CEPROBI 8, Yautepec, Morelos C.P. 62731, Mexico; iselaventura@yahoo.com.mx; 3Facultad de Estudios Superiores, Universidad Nacional Autónoma de México, Campus 4, Cuautitlán Izcalli 54714, Mexico; mcjcrg@gmail.com; 4Instituto Politécnico Nacional, Centro de Desarrollo de Productos Bióticos, Carretera Yautepec-Jojutla, km. 6.8, San Isidro, CEPROBI 8, Yautepec, Morelos C.P. 62731, Mexico; monibrisa@hotmail.com; 5Escuela de Estudios Superiores Xalostoc, Universidad Autónoma del Estado de Morelos (UAEM), Av. Nicolás Bravo s/n, Parque Industrial Cuautla, Xalostoc, Morelos C.P. 62717, Mexico; dagoguillen@yahoo.com

**Keywords:** *Ficus carica*, post-harvest rot, semi-commercial, nanoparticles, natural compounds

## Abstract

Biodegradable alternatives for the control of *Aspergillus flavus* in fig fruit were tested with the application of coatings based on chitosan (CS) and propolis (P). To potentiate the fungicidal effect, nanoparticles of these two (CSNPs and PNPs) were also considered. The objectives of this research were to evaluate the effect of different formulations on: (a) the ripening process of the fig, (b) the incidence of *A. flavus* and the production of aflatoxins, and (c) the acceptance of the treated fruit by a panel. The nanostructured coatings did not influence the ripening process of the fruit during the 12 days of storage, however, the antioxidant activity increased by approximately 30% with the coating CS + PNPs + P. The figs treated with CS + CSNPs + PNPs + P, inhibited the growth of the fungus by about 20% to 30% under laboratory and semi-commercial conditions. For all treatments, the aflatoxin production was lower than 20 ppb compared to the control with values of c.a. 250 ppb. The sensory quality was acceptable among the panel. The edible coatings can be a non-toxic alternative for post-harvest preservation and the consumption of fig fruit. The next step will be its inclusion and evaluation at a commercial level in packing houses.

## 1. Introduction

Fig fruits are rich in fibre, anthocyanins, carotenoids, flavonoids, and phenolic compounds with antioxidant properties [1,2], but during their post-harvest stage, they can be infected by the fungus *Aspergillus flavus* which affects the quality of the fruit and produces aflatoxins that are harmful to human health [3]. The control of this fungus has included the use of fungicides, yet they have generated fungal resistance. Moreover, they are detrimental to the environment and human health [4,5].

Previous researchers have shown that edible coatings based on natural, biodegradable, and non-toxic compounds such as chitosan (CS) and propolis (P) can provide antimicrobial activity against a wide variety of phytopathogenic fungi, including the genus *Aspergillus* [6,7]. To potentiate the reactivity of these compounds, it has been proposed to take advantage of nanotechnology [8]. In recent years, the benefits of using nanocoatings have been demonstrated to reduce the incidence of post-harvest pathogens and extend the storage life of different horticultural commodities including, among others, avocado, tomato, bell pepper, and strawberry [9,10,11,12].

Moreover, in previous studies carried out by Cortes-Higareda et al. [7] the nanostructured coatings based on chitosan at 0.05% and propolis at 40% notably inhibited the mycelial growth of *A. flavus* by up to 30% and, most importantly, the production of aflatoxins by <2.8 ppb. However, before further recommendation, it was necessary to evaluate the effect of these coatings on the storage life of fresh figs. Therefore, this research aimed to evaluate the chitosan/propolis nanostructured coatings during the ripening process of the fig, the development of *A. flavus* and aflatoxin production, and the sensory quality in the course of a given storage period in the laboratory and semi-commercial levels.

## 2. Materials and Methods

### 2.1. Fungal Strain

The *A. flavus* fungus was obtained from the fungal collection of the Laboratory of Post-harvest Technology of Agricultural Products at CEPROBI-IPN. The strain was activated in fig fruit and incubated in Czapeck-dox agar medium at 20 °C.

### 2.2. Materials

The fig ‘Black mission’ was obtained from an orchard located in Axochiapan, Mexico, (18°30′09″ N 98°45′14″ W). The harvest was carried out when the fruit was 75% ripe. Fruit with physical damage, irregular shapes, and the presence of microorganisms were discarded. Medium and low molecular weight chitosan Sigma Aldrich^®^ (St. Louis, MO, USA. deacetylation degree 75–85%), was used for the formulations and the synthesis of nanoparticles, respectively. The extract of propolis was obtained from Remedios Herbolarios Rosa Elena Dueñas, S.A de C. V. Glacial acetic acid and methanol were purchased from Fermont Chemicals Inc., and glycerol from J.T. Baker^®^.

### 2.3. Chitosan and Propolis Preparation

Medium molecular weight chitosan concentrations of 1.0% was prepared by adding an equal amount (*w*/*v* 1:1) of acetic acid to chitosan. The chitosan-acetic acid mixture was added to the total volume of distilled water [7]. This was stirred for 24 h at room temperature. The solution was adjusted to pH 5.5 with a 1N NaOH solution. The propolis extract was used at a concentration of 10% [9].

### 2.4. Chitosan and Propolis Nanoparticles Preparation

The chitosan and propolis nanoparticles were synthetized according to the methodology proposed by Cortes-Higareda et al. [7]. Low molecular chitosan solutions at concentrations of 0.05% (*w*/*v*) were dissolved in glacial acetic acid (1% *v*/*v*) and distilled water. 2.5 mL of the previously dissolved chitosan solution were added to a methanol solution (40 mL), by using a peristaltic pump (Bio-Rad, EP-1 Econo Pump) under moderate stirring. The solution obtained was placed in a rotary evaporator (Rotary Evaporator RE 300, BM 500 Water Bath, Yamato CF 300) at 40 °C and 50 rpm. The final volume of nanoparticles was 2 mL.

For the propolis nanoparticles (PNPs), a liquid extract of propolis at 10% was dissolved in methanol (40%) to obtain a final concentration of 0.6%. A similar methodology was used for the preparation of the chitosan nanoparticles (CSNPs).

### 2.5. Formulations and Application of Coatings

Seven coatings were prepared: (1) CS + PNPs, (2) CS + CSNPs, (3) CS + P, (4) CS + CSNPs + PNPs, (5) CS + PNPs + CSNPs + P, (6) CS + PNPs + P, (7) CS + CSNPs + P, and (8) CS. The control consisted of dipping the fig fruit in water. For the chitosan preparation, glycerol at 0.3% (*v*/*v*) was added and the propolis extract was added by dripping using a peristaltic pump. The formulation was kept at 40 °C under constant stirring for 10 min and allowed to cool at room temperature. CSNPs and PNPs were added to the formulation and stirring continued for another 5 min. The solution was then homogenized at 10,000 rpm × 1 min.

The figs were quickly washed with running water to remove excess dirt or debris and allowed to dry. Next, they were immersed for 2 min in each formulation, dried at room temperature, and at 4 ± 1 °C for 12 days [13].

### 2.6. Determination of Weight Loss, Firmness, Total Soluble Solids (TSS), Color, Respiration, and Ethylene

The fruit were weighed daily. Weight loss was determined by gravimetry with the help of a scale (OHAUS, Tokyo, Japan) which involved calculating the difference between the initial and final weight of each experimental unit, dividing this by the initial weight, and then multiplying the outcome by 100. The result was expressed as a percentage (%). The firmness was determined using an analogous penetrometer (KANDPI, Tokyo, Japan). A cylindrical tip, 8 mm in diameter, was used and both sides of the fruit were penetrated. Firmness was assessed at the beginning and end of the experiment. The values were reported as the force required to cross the membrane of the fruit in Newtons (N). To determine the total soluble solids (TSS), a drop of juice was extracted and analysed in a refractometer (Atago, Tokyo, Japan). The results were expressed in °Brix. The colour of the fruit was determined daily using a colorimeter (Konica, Tokyo, Japan). Colour values were reported in terms of the coordinate’s chromaticity ($C*=\sqrt{\left(a*\right)2+\left(b*\right)2}$) [14,15]. The fruit was placed in hermetically sealed containers at 25 ± 3 °C. After 2 h, 1 mL of the gas contained in the free space of the container was taken and injected into a gas chromatograph model 7890B GC (Agilent Technologies, California, USA), equipped with two columns: HP-PLOT/Q and CP-MOLSIEVE 5 A (Agilent Technologies, USA) and coupled to a flame ionization detector (FID) at 300 °C and thermal conductivity (TCD) at 250 °C. The injector was maintained at 200 °C in split mode 1:10, using helium as a carrier gas at a flow of 10 mL min^−1^. The results were expressed as mg of CO_2_ Kg^−1^ h^−1^ and as mg of ethylene Kg^−1^ h^−1^. Ten fruit were used per treatment in each of the variables [11].

### 2.7. Antioxidant Capacity

2,2-Diphenyl-1-picrilhidrazilo (DPPH) weighing 0.01 g was added to 25 mL with methanol (J.T. Baker). A total of 8.7 mL with methanol was then added to 1.3 mL of the DPPH solution. A fig sample of 0.5 g was weighed; next, 6 mL of methanol was added, macerated with a ceramic mortar, and centrifuged (Labnet International, USA) at 8000 rpm for 10 min. A sample of 250 μL was then taken and added to 750 μL of DPPH (133 µM). For the blank, 750 µL of DPPH was added to 250 µL of methanol. The sample was incubated in the dark at room temperature for 30 min. Absorbance at 517 nm was then measured (Thermo scientific Genesys, China) [9]. Fifteen fruit were used per treatment. Radical uptake activity was expressed as a percentage of DPPH inhibition and was calculated according to the following formula:$$\%\;\mathrm{reduction}\;\mathrm{of}\;\mathrm{DPPH}\;=\;\frac{\mathrm{Abs}0*-\mathrm{Absm}**}{\mathrm{Abs}0}\;\times \;100$$
*—Abs0 = blank absorbance and **—Absm = sample absorbance.

### 2.8. Evaluation of the Antifungal Activity in Laboratory and Semi-Commercial Conditions

The fig fruit were quickly washed with a solution of distilled water and 1% sodium hypochlorite. They were subsequently rinsed with sterile distilled water and dried at room temperature (27 ± 2 °C) on trays with absorbent paper. For evaluations at the laboratory level, 10 fruit were used per treatment, which were submerged in all treatments. The fruit were immersed in the formulations for 30 s and dried at room temperature. A wound was made on both sides and 10 µL of a spore solution of *A. flavus* (10^4^) was added; the fruit were stored at 4 °C for 12 and 15 days. The experiment was carried out in triplicate.

For the evaluation at the semi-commercial level, 66 fruit were used per treatment and only the treatments that presented an incidence equal to or less than 30% were evaluated in the laboratory experiments (CS + PNPs, CS + CSNPs, CS +CSNPs + PNPs + P, and CS + CSNPs + P). In both evaluations disease incidence caused by *A. flavus* was evaluated.

### 2.9. Aflatoxin Production in Fig Fruit Inoculated with A. flavus

The aflatoxin analysis was carried out at the laboratory of Alimentos, Micotoxinas and Micotoxicosis of the Faculty of High Studies of Cuautitlán of the National Autonomous University. The fig fruit were covered with the 8 treatments and the control. Later, the fig fruit were inoculated and stored as previously described; 10 fruit were used per treatment. Ten grams of fig fruit were liquefied with 50 mL of a methanol-water solution (70/30) for 1 min. The mixture was filtered through Whatman grade 1 paper for subsequent centrifugation at 5000 rpm for 10 min. An extract of 5 mL was deposited in an Erlenmeyer flask and 20 mL of double distilled water was added. The dilution was filtered again through Whatman grade 1 paper and 10 mL was passed through the monoclonal affinity column (Aflatest, VICAM). Subsequently, the column was rinsed with double distilled water; 1 mL of high-performance liquid chromatography (HPLC) grade methanol was passed through the column and recovered for introduction to a multiple wavelength fluorescence detector (VICAM, Series-4) using 365 nm excitation and 440 nm emission for aflatoxin detection. The data were expressed in parts per billion (ppb) of total aflatoxins [7].

### 2.10. Sensory Evaluation

The sensory evaluation was carried out on figs covered with five different treatments. Five figs were used per treatment, including the control. Covered figs were cut in half and those coded with random digits were then placed in white plastic cups. The glasses were closed for 20 min. Thirty untrained judges used two evaluation sheets to assess two random samples in which aroma, colour, and flavour were rated on a scale from 1 to 9, where 1 means “I dislike it very much” and 9 “I like it very much”.

### 2.11. Statistical Analysis

An analysis of variance (ANOVA) and a Tukey means test (*p* ≤ 0.05) were then performed, using the statistical package InfoStat student version 2017.

## 3. Results and Discussion

Overall, treated and untreated fruit gradually lost weight during the storage of 12 days at 4 °C. However, the figs coated with CS showed the lowest weight loss (25.4%) at the end of storage, being statistically (*p* ≤ 0.05) different from the remaining treatments (Figure 1). These results agree with data of Contreras-Oliva et al. [16] who evaluated edible coatings based on CS in oranges and obtained 10% lower weight loss compared to the untreated ones. Similarly, Martinez-Gonzalez et al. [9] found that in strawberries treated with edible coatings based on CS, CSNPs and P, a lower weight loss was observed (9.7%) compared to the control (14.9%). Also, Bautista-Baños et al. [14] evaluated coatings based on CS (1%), oleic acid (0.1%) and lime essential oil (0.1%) in tomato ‘Saladette’, and fruit weight loss was about 7.0%, while in the untreated fruit the value was nearer to 9.0%. Generally, weight changes are mainly due to the elimination of water caused by the transpiration processes of the fruit. However, chitosan forms a semi-permeable barrier that regulates gas exchange and reduces water loss through perspiration [17]. In addition, the glycerol added to the formulation has the ability to form hydrogen bonds and polymers, which improve flexibility and functionality, and decrease the rigidity and brittleness of the coating; this allows it to act as a plasticizing agent and, therefore, provide better barrier capacity [18].

Table 1 shows the firmness values of the treated and untreated figs during the 12 days storage period. Overall, data was statistically similar among each of the storage days, however, in the treated fruit, there was a slight tendency for higher firmness. Hernandez Lopez et al. [19] evaluated chitosan and α-pinene nanoparticle coatings in bell peppers, and argue that the application of nanostructured coatings in fruits does not statistically modify the firmness values during storage. In our study, although no statistical differences were observed between the treatments, there was a slight trend towards greater firmness in figs covered with NPs and P. On this subject, Siripatrawan and Vitchayakitti [18] reported that the addition of propolis (2%) in chitosan films reduced the permeability of water vapour in the films (0.5 g mm Pa^−1^ d^−1^ m^2^). Similarly, Bodoni et al. [20] highlighted that the incorporation of extracts of propolis (5%) into the films reduced the permeability of water vapour (2.4 g mm/h cm^2^ Pa, respectively). The propolis is composed of more than 150 substances, among them: resin, pollen, waxes, balsams, aromatic oils, and phenolic compounds [21]. The latter (polyphenolic compounds) binds to the chitosan matrix and interacts with hydrogen or covalent bonds, which limits the availability of the hydrogen atom necessary to form a hydrophilic bond with water, thus reducing the permeability to water vapour from the coatings [18].

With respect to the TSS, there were no statistically significant differences among treatments (Table 2), which is in line with research on coated tomato fruit carried out by Ramos-García et al. [22]. They reported that there were no differences in the TSS on fruit with chitosan coatings and lime essential oil with respect to the non-treated tomato. Likewise, Barrera et al. [23] and Pastor et al. [24] incorporated propolis extract into the coatings and applied them on papaya and grape cv. Moscatel fruit respectively, with no differences in values of TSS at the end of the given storage periods.

In this research, no statistical differences were observed among treatments and control in the variable colour (Table 3). The same findings were reported by Baldoni et al. [13], with the same cv. Black mission was treated with CS coatings and essential oils. Overall, changes in fruit colour are due to the ripening process, since the degradation of chlorophyll occurs and the synthesis of pigments such as anthocyanins begins [25], which is stable at a pH range between 3 and 8 in fig fruit [26]. The application of coatings with pH 5.6, as in this case, did not affect the anthocyanin content, therefore, there were no changes in colour. Colour is one of the most important sensory attributes in fruit, hence, it is expected that this variable does not change with the use of conservation technologies.

With respect to the antioxidant capacity, a pattern was not detected associated with the formulations applied or storage days. The fruit that showed the highest antioxidant capacity were treated with CS + CSNPs + PNPs (67.7 DPPH%), CS + PNPs + P (73.6 DPPH%), and CS + CSNPs + P (83.3 DPPH%) (Table 4). As reported by Solomon et al. [2], by itself, the fig contains high antioxidant capacity. The authors emphasize that this type of fruit contains biologically active compounds with antioxidant power, among which are: carotenoids, anthocyanins, flavonoids, and phenolic compounds such as gallic, chlorogenic, and cinnamic acids. On the other hand, Vargas-Sanchez et al. [27] and Navarro-Navarro [28] highlighted that propolis is a substance with great antioxidant power, due to its high content of low molecular weight phenolic compounds, with hydroxyl groups in their chemical structure, which have a sequestering mechanism of free radicals and metal ion chelation. Along the same lines, edible coatings have the ability to protect horticultural products from oxidative damage without affecting the chemical composition of the food [29] and if compounds with a high antioxidant value such as propolis are also added, the antioxidant capacity in the fruit increases and is preserved.

The fruit that showed the lowest concentration of CO_2_ and ethylene production were those treated with CS + CSNPs (71.3 mL CO_2_ kg^−1^ h^−1^ and 21.3 mL Kg^−1^ h^−1^, respectively) (Figure 2). These results agree with the study carried out by Pilon et al. [30]. They outlined that the formulations based on CSNPs applied to minimally processed apples reduced CO_2_ production and ethylene synthesis by 1.78 mL L^−1^ and 0.24 μg L^−1^, respectively, compared to the control (2.47 mL L^−1^ and 0.31 μg L^−1^, respectively). Similarly, Bautista-Baños et al. [14] reported that CS coatings with stage essential oil on tomato fruit reduced ethylene production compared to the uncoated ones. Fruit respiration is a natural process in which plant cells continue to be metabolically active and obtain the necessary energy through the catabolism of their organic reserves; these reserves are depleting, generating a change in their flavour. When applying coatings with chitosan, a semi-permeable barrier is presented on the surface of the fruit, generating a modified atmosphere regulating gas exchange without affecting the respiration rate and metabolism of the fruit [31,32].

Regarding the infection levels of the treated fruit, the coatings CS + CSNPs provided the lowest disease incidence (20% and 30%) under laboratory and semi-commercial conditions (Figure 3a,b). These data agree with the findings reported by Correa-Pacheco et al. [12]. In that study, the *Colletotrichum gloeosporioides* incidence on avocado fruit cv. Hass was reduced by up to 60% with the formulation based on CNPs + thyme essential oil. In the same line, Locaso and del Carmen [33] achieved a reduction of 55% of *Penicillium digitatum* infection on orange fruit with 24 different chitosan-based films, and Ramos-García et al. [22] reported a complete inhibition of *Escherichia coli* DH5a in tomato fruit treated with CS + beeswax + lime essential oil. On this subject, Bautista-Baños et al. [34] stated that chitosan is a polymer with confirmed antimicrobial properties in various fruit and vegetables. These authors maintain that the polycationic nature and the length of the chitosan polymer chain causes an imbalance in the ionic cellular homeostasis of K and Ca^2^ when joining with the membrane of the microorganisms, provoking, among other physiological alterations, the exit of various molecules that involve the nutrition of the microorganism [6,35].

As for the aflatoxin content, all the coated figs, regardless of the formulation applied, had values of <20 ppb, while the non-treated figs presented the highest quantity of aflatoxins of c.a. 250 ppb (Figure 4). Cortes-Higareda et al. [7] reported similar results, but in this case, the study was carried out in vitro. *Aspergillus flavus* aflatoxins on a formulation based on CS formulation was about 2.80 ppb. Similarly, Torlak and Sert [36] succeeded in inhibiting the production of aflatoxins 100% in in vitro with nanostructured coatings based on chitosan and propolis (40%). The authors pointed out that this could be explained by the synergistic activity of the compounds and the advantages that the nanostructuring provides. Chitosan has the ability to adsorb aflatoxin B1 when the positive charges of its amino group interact with the negative charges of the oxygen atoms of aflatoxins [37]. Therefore, the ability of chitosan to inhibit aflatoxins is a mechanism that involves ion chelation, for this it requires the –OH and –O groups of D-glucosamine residues as binders, and 2 or more amino groups of the same chain to join the same ion [38].

Regarding sensory quality, the treatments were significantly similar to the control in the variables of appearance, aroma and flavour (Table 5). These findings agree with the study of Martinez-Gonzalez et al. [9]. The researchers evaluated CS, CSNPs, PNPs, and P nanostructured coatings in strawberry fruit and did not observe differences in the variables taste, appearance and colour; they also reported any modification in the odour.

## 4. Conclusions

The figs treated with the coatings maintained their normal ripening process during the 12-day storage, and were subject to reduced weight loss; furthermore, antioxidant capacity increased. Regarding the development of *A. flavus*, the coatings reduced the presence of this fungus under controlled laboratory and semi-commercial conditions, and aflatoxin production reduced remarkably. The consumer acceptance of the treated fruit was normal. For this reason, the tested coatings could be a viable, safe, and non-toxic alternative for the post-harvest preservation and consumption of fig fruit. The next step will be its inclusion and evaluation at a commercial level in packing houses.

## Figures and Tables

**Figure 1 plants-10-00112-f001:**
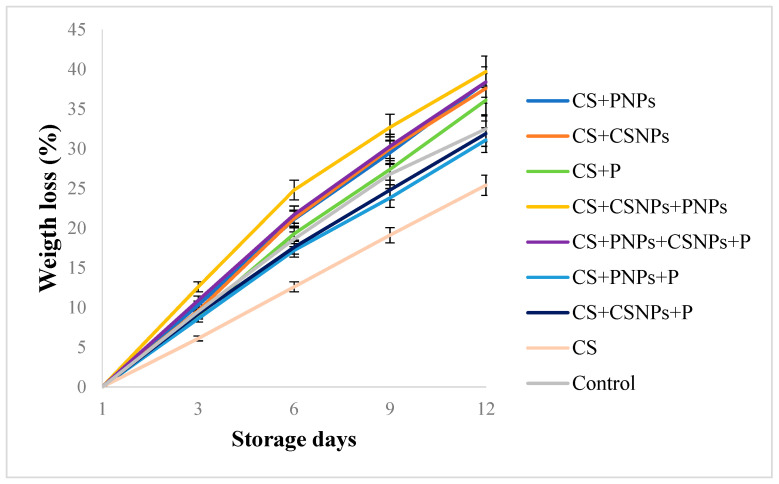
Effect of edible chitosan/propolis coatings on the weight loss of fig fruit stored for 12 days at 4 °C. CS (chitosan), P (propolis extract), CSNPs (chitosan nanoparticles), PNPs (propolis nanoparticles).

**Figure 2 plants-10-00112-f002:**
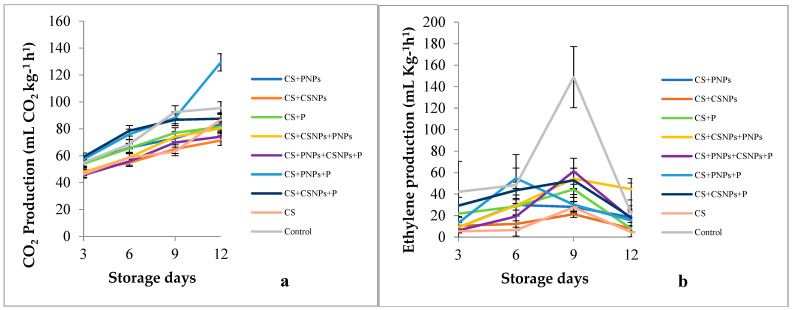
Effect of edible of chitosan/propolis coatings on: (**a**) CO_2_ and (**b**) ethylene production of fig fruit stored for 12 days at 4 °C. CS (chitosan), P (propolis extract), CSNPs (chitosan nanoparticles), PNPs (propolis nanoparticles).

**Figure 3 plants-10-00112-f003:**
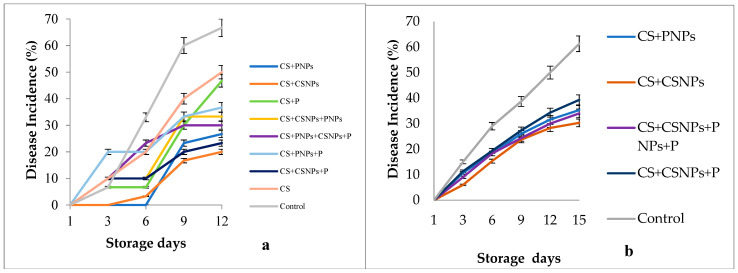
Effect of edible chitosan/propolis coatings on the incidence of *A. flavus* in fig fruit stored for 12 days at 4 °C under: (**a**) laboratory and (**b**) semi-commercial conditions. CS (chitosan), P (propolis extract), CSNPs (chitosan nanoparticles), PNPs (propolis nanoparticles).

**Figure 4 plants-10-00112-f004:**
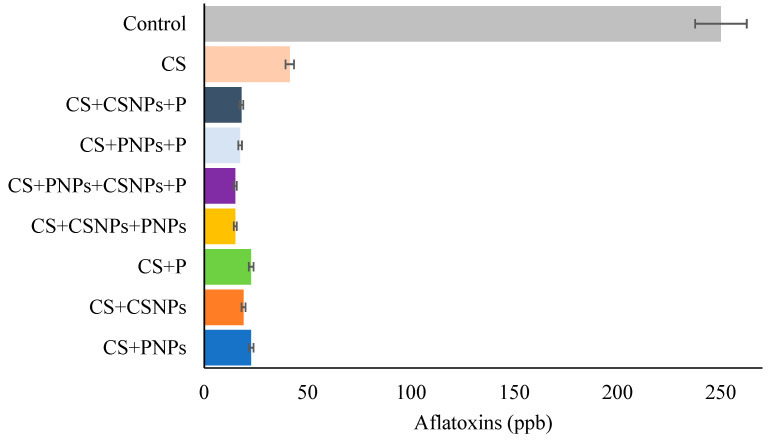
Effect of edible of chitosan/propolis on the aflatoxins production of *A. flavus* in fig fruit stored for 15 days at 4 °C.

**Table 1 plants-10-00112-t001:** Effect of edible chitosan/propolis coatings on the firmness of fig fruit stored for 12 days at 4 °C.

	Firmness (N)				
Coatings	Storage Days				
	1	3	6	9	12
CS + PNPs	5.6 ± 1.4 ^a^	3.1 ± 1.1 ^a^	5.4 ± 1.4 ^ab^	6.2 ± 2.1 ^a^	5.1 ± 1.3 ^b^
CS + CSNPs	6.3 ± 1.7 ^a^	4.6 ± 3.0 ^ab^	7.0 ± 1.0 ^b^	5.8 ± 2.0 ^a^	4.5 ± 1.1 ^ab^
CS + P	7.0 ± 0.8 ^a^	4.3 ± 1.3 ^ab^	5.3 ± 1.9 ^ab^	5.6 ± 1.5 ^a^	2.8 ± 1.1 ^a^
CS + CSNPs + PNPs	4.5 ± 1.4 ^a^	6.0 ± 1.1 ^ab^	4.6 ± 1.7 ^ab^	5.3 ± 0.9 ^a^	6.0 ± 1.3 ^b^
CS + PNPs + CSNPs + P	6.1 ± 1.7 ^a^	6.2 ± 0.4 ^ab^	6.1 ± 1.6 ^ab^	4.4 ± 0.9 ^a^	5.6 ± 1.0 ^b^
CS + PNPs + P	4.8 ± 1.2 ^a^	4.9 ± 1.2 ^ab^	5.0 ± 1.0 ^ab^	4.4 ± 0.8 ^a^	4.4 ± 1.0 ^ab^
CS + CSNPs + P	5.5 ± 1.2 ^a^	5.8 ± 2.2 ^ab^	6.2 ± 1.0 ^ab^	4.9 ± 0.4 ^a^	5.6 ± 0.5 ^b^
CS	5.0 ± 1.7 ^a^	3.9 ± 0.5 ^ab^	3.9 ± 0.9 ^a^	3.8 ± 1.2 ^a^	4.1 ± 0.7 ^ab^
Control	6.0 ± 1.6 ^a^	6.4 ± 1.1 ^b^	4.6 ± 0.9 ^ab^	4.4 ± 0.6 ^a^	4.9 ± 0.6 ^b^

Means with similar letters are not significantly different among the evaluated treatments. CS (chitosan), CSNPs (chitosan nanoparticles), PNPs (propolis nanoparticles), P (propolis extract). Analysis of variance (ANOVA) and Tukey test (*p* ≤ 0.05) were performed.

**Table 2 plants-10-00112-t002:** Effect of edible chitosan/propolis coatings on total soluble solids (TSS) concentration of fig fruit stored for 12 days at 4 °C.

	TSS (°BRIX)				
Coatings	Storage Days				
	1	3	6	9	12
CS + PNPs	16.2 ± 1.9 ^ab^	20.6 ± 3.8 ^abc^	20.8 ± 1.8 ^a^	23.0 ± 3.7 ^ab^	22.2 ± 4.0 ^a^
CS + CSNPs	14.8 ± 2.9 ^a^*	20.8 ± 3.6 ^abc^	18.2 ± 2.4 ^a^	20.6 ± 3.8 ^ab^	21.2 ± 2.6 ^a^
CS + P	17.4 ± 3.4 ^ab^	19.8 ± 2.8 ^abc^	16.4 ± 5.0 ^a^	26.6 ± 0.6 ^b^	24.0 ± 3.9 ^a^
CS + CSNPs + PNPs	18.2 ± 2.9 ^ab^	17.4 ± 2.9 ^ab^	15.8 ± 2.2 ^a^	19.8 ± 3.4 ^ab^	24.0 ± 3.7 ^a^
CS + PNPs + CSNPs + P	16.6 ± 2.7 ^ab^	17.6 ± 4.5 ^ab^	19.2 ± 3.4 ^a^	22.4 ± 4.7 ^ab^	22.8 ± 5.5 ^a^
CS + PNPs + P	21.2 ± 2.4 ^b^	23.0 ± 3.4 ^abc^	21.4 ± 2.6 ^a^	20.4 ± 3.5 ^ab^	25.6 ± 2.1 ^a^
CS + CSNPs + P	21.2 ± 2.8 ^b^	24.8 ± 2.6 ^c^	19.6 ± 4.5 ^a^	21.8 ± 3.7 ^ab^	22.6 ± 2.8 ^a^
CS	21 ± 3.6 ^b^	23.6 ± 2.5 ^bc^	19.4 ± 2.7 ^a^	18.4 ± 2.88 ^a^	20.0 ± 4.5 ^a^
Control	18 ± 1.9 ^ab^	16.8 ± 2.8 ^a^	18.8 ± 2.4 ^a^	23.0 ± 2.1 ^ab^	22.0 ± 5.7 ^a^

* Means with similar letters are not significantly different among the evaluated treatments. CS (chitosan), P (propolis extract), CSNPs (chitosan nanoparticles), PNPs (propolis nanoparticles). ANOVA and Tukey test (*p* ≤ 0.05) were performed.

**Table 3 plants-10-00112-t003:** Effect of edible chitosan/propolis coatings on the colour (chromaticity) of fig fruit stored for 12 days at 4 °C.

	CHROMATICITY (C*)				
Coatings	Storage Days				
	1	3	6	9	12
CS + PNPs	8.3 ± 2.6 ^a^	6.4 ± 2.3 ^a^	4.5 ± 2.0 ^a^	4.7 ± 2.4 ^a^	3.7 ± 2.2 ^a^
CS + CSNPs	7.2 ± 2.6 ^a^	5.5 ± 1.5 ^a^	4.2 ± 1.2 ^a^	3.9 ± 1.8 ª	3.7 ± 1.5 ^a^
CS + P	7.5 ± 3.5 ^a^	7.4 ± 2.5 ^a^	5.8 ± 2.3 ^a^	4.8 ± 1.7 ª	3.9 ±1.3 ^a^
CS + CSNPs + PNPs	6.8 ± 2.8 ^a^	6.2 ± 2.9 ^a^	4.8 ± 2.6 ^a^	4.1 ± 2.4 ª	3.4 ± 2.3 ^a^
CS + PNPs + CSNPs + P	7.7 ± 4.1 ^a^	5.5 ± 2.3 ^a^	4.1 ± 2.1 ^a^	2.9 ± 1.2 ª	2.7 ± 1.8 ^a^
CS + PNPs + P	6.5 ± 3.0 ^a^	6.3 ± 3.1 ª	4.7 ± 2.4 ^a^	4.0 ± 2.0 ^a^	3.2 ± 1.4 ^a^
CS + CSNPs + P	8.2 ± 5.0 ^a^	7.6 ± 4.2 ^a^	4.8 ± 2.3 ^a^	4.5 ± 2.7 ^a^	3.7 ± 2.1 ^a^
CS	6.7 ± 2.4 ^a^	6.9 ± 3.3 ^a^	4.8 ± 1.3 ^a^	4.2 ± 1.3 ^a^	3.3 ± 1.7 ^a^
Control	7.7 ± 3.7 ^a^	6.9 ± 3.7 ^a^	5.3 ± 3.1 ^a^	3.8 ± 2.7 ^a^	3.5 ± 2.1 ^a^

Means with similar letters are not significantly different among the evaluated treatments. CS (chitosan), P (propolis extract), CSNPs (chitosan nanoparticles), PNPs (propolis nanoparticles). ANOVA and Tukey test (*p* ≤ 0.05) were performed.

**Table 4 plants-10-00112-t004:** Effect of edible chitosan/propolis coatings on the antioxidant capacity of fig fruit stored for 12 days at 4 °C.

Coatings	Antioxidant Capacity (DPPH %) Storage Days				
	1	3	6	9	12
CS + PNPs	36.4 ± 10.0 ^abc^*	38.1 ± 6.8 ^ab^*	54.8 ± 13.0 ^a^	39.0 ± 24.3 ^ab^	48.8 ± 3.0 ^ab^
CS + CSNPs	29.3 ± 3.1 ^ab^	49.0 ± 10.1 ^abc^	57.6 ± 16.9 ^a^	38.1 ± 9.6 ^ab^	43.0 ± 18.0 ^ab^
CS + P	42.5 ± 26.2 ^abc^	32.9 ± 16.6 ^a^	47.7 ± 3.3 ^a^	32.8 ± 3.0 ^a^	60.7 ± 12.6 ^bc^
CS + CSNPs + PNPs	56.0 ± 7.3 ^c^	34.9 ± 6.1 ^ab^	56.6 ± 11.3 ^a^	67.7 ± 15.6 ^cd^	62.7 ± 7.1 ^bc^
CS + PNPs + CSNPs + P	40.8 ± 11.4 ^abc^	61.0 ± 9.7 ^c^	54.6 ± 6.9 ^a^	48.7 ± 12.8 ^abc^	61.1 ± 9.1 ^bc^
CS + PNPs + P	28.1 ± 9.4 ^a^	44.8 ± 11.7 ^abc^	63.9 ± 11.6 ^a^	60.6 ± 10.7 ^bcd^	73.6 ± 17.4 ^c^
CS + CSNPs + P	50.4 ± 6.2 ^bc^	49.4 ± 2.7 ^abc^	52.9 ± 10.7 ^a^	83.3 ± 2.8 ^d^	43.6 ± 9.8 ^ab^
CS	54.5 ± 11.9 ^c^	50.9 ± 5.1 ^bc^	47.4 ± 25.4 ^a^	47.7 ± 7.1 ^abc^	37.6 ± 9.2 ^a^
Control	45.0 ± 4.4 ^abc^	44.9 ± 3.1 ^abc^	52.0 ± 6.8 ^a^	53.8 ± 7.9 ^abc^	57.0 ± 14.8 ^ab^

* Means with similar letters are not significantly different among the evaluated treatments. CS (chitosan), P (propolis extract), CSNPs (chitosan nanoparticles), PNPs (propolis nanoparticles). DPPH (2,2-Diphenyl-1-picrilhidrazilo, this free radical is capable of reacting with antioxidant compounds through a process characterized by the transfer of a hydrogen atom provided by the antioxidant agent). ANOVA and Tukey test (*p* ≤ 0.05) were performed.

**Table 5 plants-10-00112-t005:** Sensory characteristics of fig fruit treated with different nanostructured coatings.

Treatment	Appearance	Smell	Flavour
CS + PNPs	8.6 ± 0.5 ^b^	6.8 ± 1.4 ^ab^	6.6 ± 2.5 ^a^
CS + CSNPs	7.2 ± 1.5 ^a^	6.5 ± 1.3 ^ab^	6.7 ± 1.8 ^a^
CS + PNPs + CSNPs + P	7.8 ± 0.9 ^ab^	6.7 ± 1.5 ^ab^	7.6 ± 1.8 ^a^
CS + CSNPs + P	8.3 ± 1.0 ^ab^	8.0 ± 0.5 ^b^	7.2 ± 1.6 ^a^
Control	7.3 ± 1.1 ^ab^	6.2 ± 1.0 ^a^	7.2 ± 1.7 ^a^

Means with similar letters are not significantly different among the evaluated treatments. CS (chitosan), P (propolis extract), CSNPs (chitosan nanoparticles), PNPs (propolis nanoparticles). ANOVA and Tukey test (*p* ≤ 0.05) were performed.

## Data Availability

Not applicable.

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
