# Peer review of "Edible Chitosan/Propolis Coatings and Their Effect on Ripening, Development of Aspergillus flavus, and Sensory Quality in Fig Fruit, during Controlled Storage"

_plants, 2021, doi:10.3390/plants10010112_

Round 1
Reviewer 1 Report
This is a nice paper with very useful results.
In the abstract, propolis is refereed to as a compound. It is a mixture of compounds.
Define TSS.
I see no mention of the work of Elizabeth Baldwin, one of the pioneers of edible coatings for food.
The paper was well written, simple and clear.
Author Response
Line 25
Commentary. This an erroneous tratment name- no such treatment was evaluated. Do you mean CS+PNPs+P, or CS+CSNPs+P, or both? Please specify using the correct treatment name.
Answer. The suggested change was made.
Line 73
Commentary. Can you add a citation to say why the 1% concentration was used
Answer. The suggested change was made.
Line 72-73
Commentary. Does this sentence mean that acetic acid, chitosan and water were all added together to form a 1% chitosan solutuion? The meaning is not very clear.
Answer. The indicated change was made.
Line 80
Commentary. I also find this section (lines 77-81) confusing. Do you mean that the chitosan was first dissolved in acetic acid, and then you added the methanol?.
Answer. The indicated change was made.
Line 99
Commentary. Total soluble solids (TSS) Should be spelled out in full at first use, followed by the abbreviation in parentheses.
Answer. The indicated change was made.
Line 112, 122
Commentary. Maybe include a reference which explains what these coordinates refer to
Spell out DPPH in full at first use.
Answer. The indicated change was made.
Line 124
Commentary. Lines 120-121 are confusing, as per the comments by reviewer 2.
Answer. The indicated change was made.
Line 143
Commentary. This is an extremely long sentence (lines 134-140). Talk about the laboratoty and semmi commercial evaluations in separate sentences.
Answer. The indicated change was made.
Line 146
Commentary. You need to mention how you carried out the actual disease assessment. For example Do you measure % incidence or % severety or both?
Answer. The indicated change was made.
Line 166
Commentary. Dis these volunteer judges receive any specific training in sensory evaluation?
Answer. The indicated change was made.
Line 168
Commentary. I suggest the use os “strongly” rather than “extremely”
Answer. The indicated change was made.
Line 237
Commentary. What are the actual units of colour measurement here? More information is required.
Answer. The indicated change was made.
Line 237
Commentary. Units?
Answer. The indicated change was made.
Line 261
Commentary. You need to spell this abbreviation out in the figure caption and explain very briefly how it is used to determine antioxidant capacity.
Answer. The indicated change was made.

Reviewer 2 Report
- On line 107 they say that you get “a drop of strawberry juice”. It's a mistake?
- In different places, CO2 appears per CO2, Kg per kg, cm2 per cm2 and ml per mL
- DDPH appears, the first time its meaning would have to be indicated
- The instructions to determine the antioxidant capacity are quite complicated, in my opinion. They say: “A total of 8.7 ml with methanol (?) was then added to 1.3 ml of the solution (?)”… How long does the maceration last?
- How were the five treatments used in sensory analysis selected?
- In line 202 there is a “respectively” that I do not know what it refers to
- They must take care of the precision of the figures in the tables
- There is some explanation for the oscillatory evolution of some parameters?
- Some questions appear: which treatment is the best? How much would the shelf life of figs increase?
Author Response

(The authors gave the same response as above.)
